# Effort cost of harvest affects decisions and movement vigor of marmosets during foraging

Paul Hage, In Kyu Jang, Vivian Looi, Mohammad Amin Fakharian, Simon P Orozco, Jay S Pi, Ehsan Sedaghat-Nejad, Reza Shadmehr*

Laboratory for Computational Motor Control, Department of Biomedical Engineering, Johns Hopkins School of Medicine, Baltimore, United States

**Abstract** Our decisions are guided by how we perceive the value of an option, but this evaluation also affects how we move to acquire that option. Why should economic variables such as reward and effort alter the vigor of our movements? In theory, both the option that we choose and the vigor with which we move contribute to a measure of fitness in which the objective is to maximize rewards minus efforts, divided by time. To explore this idea, we engaged marmosets in a foraging task in which on each trial they decided whether to work by making saccades to visual targets, thus accumulating food, or to harvest by licking what they had earned. We varied the effort cost of harvest by moving the food tube with respect to the mouth. Theory predicted that the subjects should respond to the increased effort costs by choosing to work longer, stockpiling food before commencing harvest, but reduce their movement vigor to conserve energy. Indeed, in response to an increased effort cost of harvest, marmosets extended their work duration, but slowed their movements. These changes in decisions and movements coincided with changes in pupil size. As the effort cost of harvest declined, work duration decreased, the pupils dilated, and the vigor of licks and saccades increased. Thus, when acquisition of reward became effortful, the pupils constricted, the decisions exhibited delayed gratification, and the movements displayed reduced vigor.

*For correspondence:
shadmehr@jhu.edu

Competing interest: The authors declare that no competing interests exist.

## eLife assessment

This **important** study unravels the interaction between effort cost, pupil-indexed brain state, and movement (saccadic) vigor during foraging decisions in marmoset monkeys. Based on a normative computational model, the authors derive the prediction that anticipated effort should affect both decisions and movement vigor during foraging; and then provide **solid** behavioral and pupillometric evidence for this prediction in a foraging task. This paper will be of interest to decision and motor neuroscience as well as to all researchers studying animal behavior.

## Introduction

During foraging, animals work to locate a food cache and then spend effort harvesting what they have found. As they forage, their decisions appear to maximize a measure that is relevant to fitness: the sum of rewards acquired, minus efforts expended, divided by time, termed the capture rate (*Charnov, 1976*; *Cowie, 1977*; *Shadmehr and Ahmed, 2020*). For example, a crow will spend effort extracting a clam from a sandy beach, but if the clam is small, it will abandon it because the additional time and effort required to extract the small reward – dropping it repeatedly from a height onto rocks – can be better spent finding a bigger prize (*Richardson and Verbeek, 1986*). In other words, if going to

the bank entails waiting in a long line, one should go infrequently, but make each transaction a large amount.

Intriguingly, reward expectation not only affects decisions, it also affects movements: we not only prefer the less effortful option, we also move vigorously to obtain it (*Yoon et al., 2020*; *Korbisch et al., 2022*). This modulation of movement vigor can be justified if we consider that movements require expenditure of time and energy, which discount the value of the promised reward (*Shadmehr et al., 2010*; *Shadmehr et al., 2016*). Thus, from a theoretical perspective, there should exist a mechanism to coordinate control of decisions with control of movements so that both contribute to maximizing fitness (*Yoon et al., 2018*).

To study this coordination, we designed a task in which marmosets decided how long to work before they harvested their food. On a given trial, they made a sequence of saccades to visual targets and received an increment of food as their reward. However, the increment was small, and its harvest was effortful, requiring them to insert their tongue inside a small tube. Theory predicted that in order to maximize the capture rate, harvest should commence only when there was sufficient reward accumulated to justify the effort required for its extraction. Indeed, the subjects chose to work and stockpile food, and only then initiated their harvest.

On some days the effort cost of harvest was low: the tube was placed close to the mouth. On other days the same amount of work, that is, saccade trials, produced food that had a higher effort cost: the tube was located farther away. The theory made two interesting predictions: as the effort cost increased, the subjects should choose to work more trials, thus delaying their harvest so to stow more food, but reduce their movement vigor, thus saving energy. Indeed, when marmosets encountered an increased effort cost, they extended their work period, stockpiling food, but reduced their vigor, slowing their saccades during the work period, and slowing their licks during the harvest period.

What might be a neural basis for this coordinated response of the decision-making and the motor-control circuits? During the work and the harvest periods, momentary changes in pupil size closely tracked the changes in vigor: pupil dilation accompanied increases in vigor, while pupil constriction accompanied decreases in vigor. Remarkably, this was true regardless of whether the movement that was being performed was a saccade or a lick. Moreover, in response to the increased effort cost, the pupils exhibited a global change, constricting during both the work and the harvest periods.

If we view the changes in pupil size as a proxy for activity in the brainstem noradrenergic circuits (*Joshi and Gold, 2020*), our results suggest that as these circuits respond to effort costs (*Bornert and Bouret, 2021*), they alter computations in the brain regions that control decisions, delaying gratification and encouraging work, and the brain regions that control movements, promoting sloth and conserving energy.

## Results

We tracked the eyes and the tongue of head-fixed marmosets as they performed visually guided saccades in exchange for food (*Figure 1A*). Each successful trial consisted of three visually guided saccades, at the end of which we delivered an increment of food (a slurry mixture of apple sauce and monkey chow). Because the reward amount was small (0.015–0.02 mL), the subjects rarely harvested following a single successful trial. Rather, they worked for a few trials, allowing the food to accumulate, then initiated their harvest by licking (*Figure 1B*). The key variables were how many trials they chose to work before starting harvest, and how vigorously they moved their eyes and tongue during the work and the harvest periods.

Over the course of 2.5 y, we recorded 56 sessions in subject M (29 mo) and 56 sessions in subject R (23 mo). A typical work period lasted about 10 s, during which the subjects attempted ~8 trials and succeeded in 4–5 trials (*Figure 1D*) (a successful trial was when all three saccades were within 1.25° of the center of each target). The work period ended when the subject decided to stop tracking the targets and instead initiated harvest, which lasted about 6 s, resulting in 16–18 licks. Subject M completed an average of 909.5 ± 61 successful trials per session (mean ± SEM), producing an average of 241 ± 13.9 work-harvest pairs, and subject R completed an average of 1431 ± 65 successful trials, producing an average of 263 ± 8.9 work-harvest pairs.

We delivered food via either the left or the right tube for 50–300 consecutive trials and then switched tubes. We tracked the motion of the tongue using DeepLabCut (*Mathis et al., 2018*), as shown for a typical session in *Figure 1B*. The licks required precision because the tube was just large

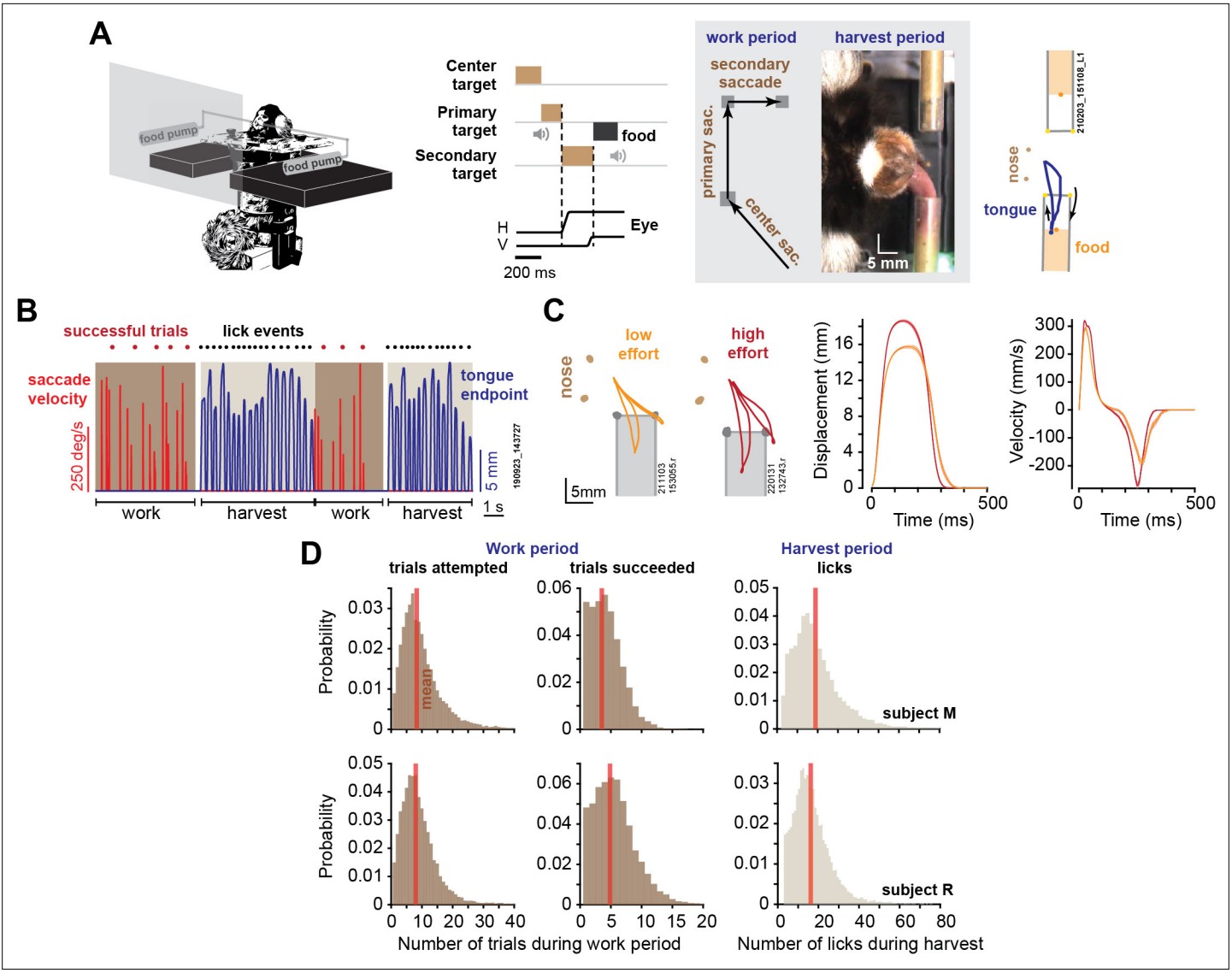

**Figure 1.** Elements of a foraging task. (**A**) During the work period, marmosets made a sequence of saccades to visual targets. A trial consisted of three consecutive saccades, at the end of which the subject was rewarded by a small increment of food. We tracked the eyes, the tongue, and the food. (**B**) An example of two consecutive work-harvest periods, showing reward-relevant saccades (eye velocity) and tongue endpoint displacement with respect to the mouth. (**C**) Data for two sessions, one where the tube was placed close to the mouth (orange trace), and one where it was placed farther away (red trace). Two types of licks are shown: inner-tube licks and outer-tube licks. Depending on food location, both types of licks can contact the food. Data on the right two panels show endpoint displacement and velocity of the tongue during inner-tube licks. Error bars are SEM. (**D**) During the work period, the subjects attempted ~8 trials on average, succeeding in 4–5 trials before starting harvest, and then licked about 18 times to extract the food.

The online version of this article includes the following figure supplement(s) for figure 1:

**Figure supplement 1.** Number of licks per harvest as a function of tube distance.

enough (4.4 mm diameter) to allow the tongue to penetrate. As a result, about 30% of the reward-seeking licks were successful and contacted food (30 ± 1.6% for subject M, 28 ± 2.5% for subject R), as shown in *Animation 1*. Examples of licks that failed to contact food are shown in *Animation 2–4*.

## Theory and predictions

During the decision-making part of the task, the brain explicitly determined how long to work before initiating harvest. During the work and the harvest periods, the brain implicitly controlled the vigor of movements. We imagined that these two forms of behavior were not independent, but rather coordinated via a control policy that maximized a single utility: the sum of rewards acquired, minus efforts

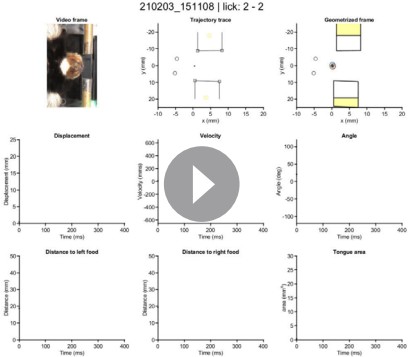

**Animation 1.** Example of a successful inner-tube lick.

**Animation 2.** Example of an under-tube lick that failed to contact food. Note the corrective sub-movements, as has been observed in mice (***Bollu et al., 2021***).

expended, divided by time, termed the capture rate. We chose this formulation because it presents a normative approach that ecologists have used to understand the decisions that animals make regarding how far to travel for food, what mode of travel to use, and how long to stay before moving on to another patch (***Richardson and Verbeek, 1986***; ***Stephens and Krebs, 1987***; ***Bautista et al., 2001***).

During a work period, our subjects decided to complete a number of saccade trials $n_s$, a fraction $\beta_s$ of which were successful, earning food increment $\alpha$, but expended effort $c_s$ that consumed time $T_s$ for each trial. They then stopped working and initiated harvest, producing a number of licks $n_l$, a fraction $\beta_l$ of which succeeded, thus expending effort $c_l$ and consuming time $T_l$ for each lick. These actions produced the following capture rate:

$$J = \frac{\alpha \beta_s n_s \left(1 - \dfrac{1}{1 + \beta_l n_l}\right) - n_l c_l - n_s^2 c_s}{n_l T_l + n_s T_s} \tag{1}$$

In the numerator of ***Equation 1***, the first term represents the fact that the food cache increased linearly with successful trials and was then consumed gradually with successful licks. The second term represents the effort expenditure of licking, and the third term represents the effort expenditure of working. Notably, the effort expenditure of work, $n_s^2 c_s$, grows faster than linearly as a function of trials. This nonlinearity is essential to reflect the idea that following a long work period, the capture rate must be more negative than following a short work period (i.e., more work trials produce a greater reduction in utility).

A control policy describes how long to work and harvest, and an optimal policy produces periods of working and harvesting, $n_s^*$, $n_l^*$, that maximize ***Equation 1***. A closed-form solution for the optimal

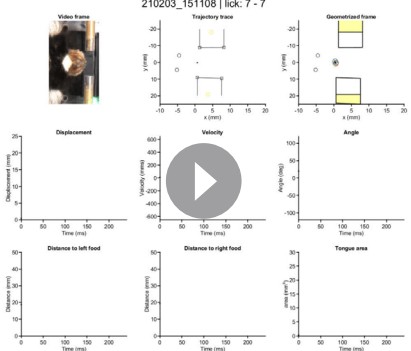

**Animation 3.** Example of an outer-tube lick that failed to contact food.

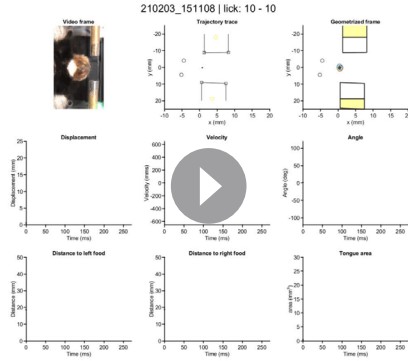

**Animation 4.** Example of a lick that hit the outer edge of the tube and failed to contact food.

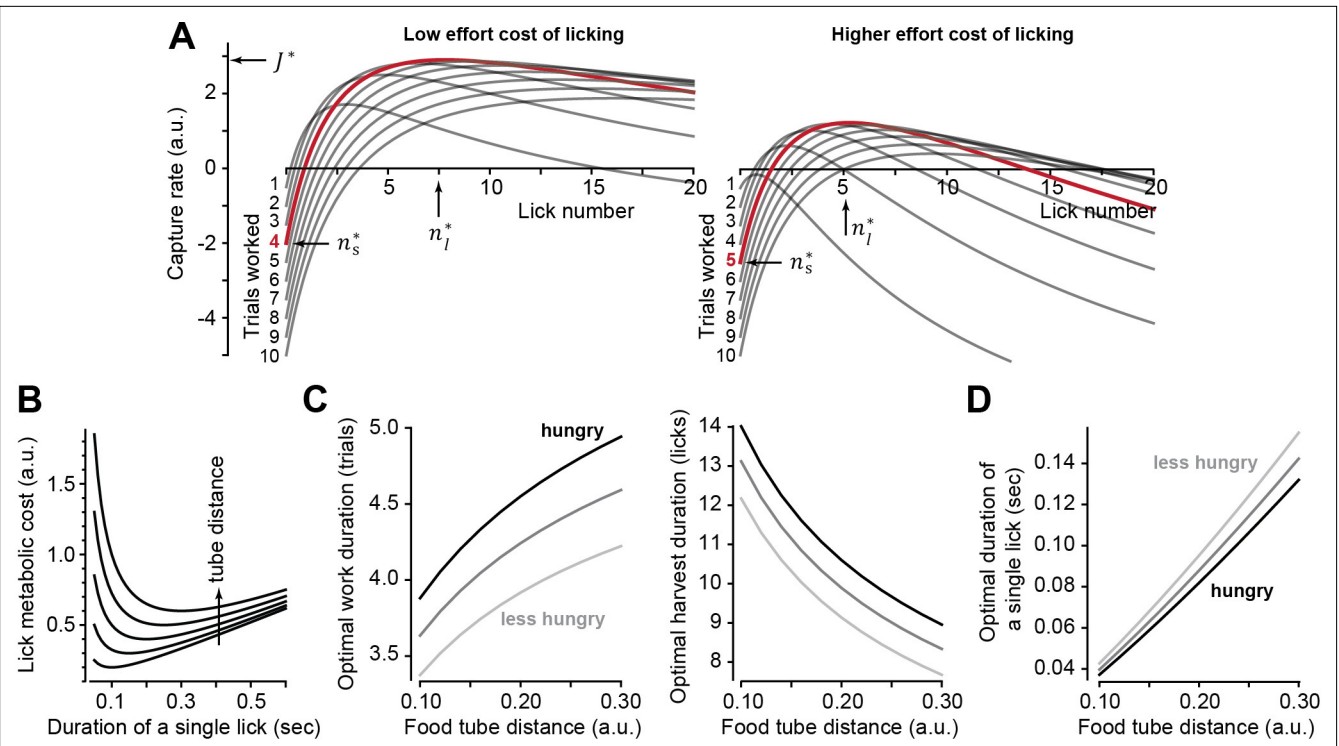

**Figure 2.** Theoretical results of an optimal control policy. (**A**) Capture rate (*Equation 1*) is plotted during the harvest period as a function of lick number $n_l$ following various number of work trials $n_s$. When the effort cost of licking is low (left plot, $c_l = 0.5$), the optimal work period is $n_s^* = 4$ (red trace). When the effort cost is higher (right plot, $c_l = 2$), it is best to work longer before initiating harvest. (**B**) The metabolic cost of licking (*Equation 2*) is minimized when a lick has a specific duration. Tube distance varied from 0.1 to 0.3. Optimal duration that minimizes lick cost grows linearly with tube distance. (**C**) Optimal number of work trials $n_s^*$ and licks $n_l^*$ as a function food tube distance $d$. As the effort cost of harvest increases, one should respond by working longer, delaying harvest. (**D**) Optimal lick duration $T_l^*$ as a function of food tube distance. The lick duration $T_l^*$ that maximizes the capture rate is smaller than the one that minimizes the lick metabolic cost (**B**). That is, it is worthwhile moving vigorously to acquire reward. However, $T_l^*$ grows faster than linearly as a function of tube distance. Thus, as the tube moves farther, it is best to reduce lick vigor. Hunger, modeled as increased value of reward, should promote work and increase vigor, while effort cost of harvest (tube distance) should promote work but reduce vigor. Parameter values for all simulations: $\beta_s = 0.5$, $\beta_l = 0.3$, $T_s = 1$, $k = 1$, $\alpha = 20$ (low food value, less hunger), $\alpha = 25$ (high food value, hungry).

policy can be obtained (*Supplementary file 1*), and *Figure 2A* provides an example. As the work period concludes and the harvest period begins ($n_l = 0$), the capture rate is negative. This reflects the fact that the subject has performed a few trials and stockpiled food, thus expended effort but has not been rewarded yet. The capture rate rises when licking commences. Critically, the peak capture rate is not an increasing function of the work period. Rather, there is an optimal work period ($n_s^*$, red trace, *Figure 2A*) associated with a given effort cost of licking $c_l$. If we now move the tube away from the mouth, that is, increase the effort cost of licking $c_l$, the peak of the capture rate shifts and the optimal work period changes: the proper response to an increased effort cost of licking is to work longer, stowing more food before commencing harvest.

Notably, the higher cost of licking inevitably reduces the maximum capture rate (*Figure 2A*). This should impact movement vigor: animals tend to respond to a reduced capture rate by slowing their movements (*Yoon et al., 2018*), which can be viewed as an effective way to save energy (*Shadmehr et al., 2016*). To incorporate vigor into the capture rate, we tried to define the effort cost of a single lick $c_l$ in terms of its energetic cost, a relationship that is currently unknown. Fortunately, other movements provide a clue: the energetic cost of reaching (*Shadmehr et al., 2016*; *Huang and Ahmed, 2014*) and the energetic cost of walking (*Ralston, 1958*; *Bastien et al., 2005*) are both concave upward functions of the movement's duration. That is, from an energetic standpoint, there is a reach speed and a walking speed that minimize the cost of each type of movement. We generalized these empirical observations to licking and assumed that the energetic cost of a single lick was a concave upward function of its duration:

$$c_l\left(T_l\right) = \frac{d^2}{T_l} + kT_l \tag{2}$$

In *Equation 2*, the lick is aimed at a tube located at distance $d$ and has a duration $T_l$. The parameter $k$ describes the rate with which the cost grows as a function of duration. For example, the lick duration that minimizes the energetic cost is $d/\sqrt{k}$. Thus, for an energetically optimal lick, duration grows linearly with tube distance (*Figure 2B*). However, our objective is not to minimize the cost of licking, but to maximize the capture rate. To do so, we insert *Equation 2* into *Equation 1* and find the optimal policy $(n_s^*, n_l^*, T_l^*)$, which now depends on the distance of the food tube to the mouth (*Supplementary file 1*).

The theory predicts that to maximize the capture rate (*Equation 1*), the response to an increased effort cost of harvest (i.e., tube distance) should be as follows: $n_s^*$ should increase (*Figure 2C*), $n_l^*$ should decrease (*Figure 2C*), and $T_l^*$ should increase (*Figure 2D*). Notably, the rate of increase in $T_l^*$ as a function of tube distance is faster than linear, while from an energetic point of view (*Equation 2*), increase in distance should produce a linear increase in lick duration. Thus, as the harvest becomes more effortful, the subject should work longer to stockpile food, but move slower to save energy.

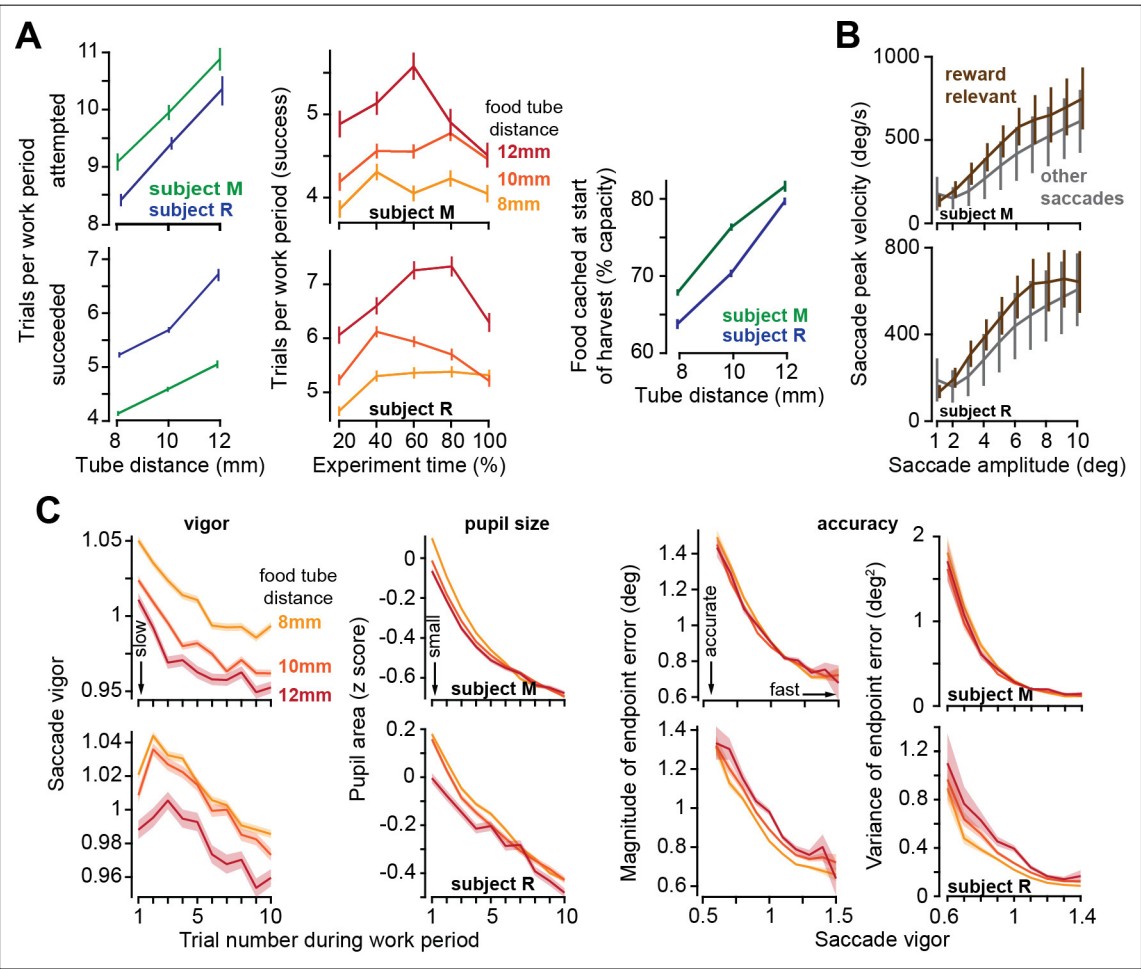

**Figure 3.** As the effort cost of harvest increased, subjects chose to work more trials, but slowed their movements. (**A**) Left: the number of trials attempted and succeeded per work period as a function of tube distance. Middle: successful trials per work period as a function of time during the recording session. Tube distance is with respect to a marker on the nose. Right: food available in the tube at the start of the harvest. (**B**) Peak saccade velocity as a function of amplitude for reward-relevant and other saccades. (**C**) Vigor of reward-relevant saccades as a function of trial number during the work period. Saccade vigor was greater when the tube was closer. Pupil size is quantified during the same work periods. Accuracy is quantified as the magnitude of the saccade's endpoint error vector (with respect to the target) and the variance of that error vector (determinant of the variance-covariance matrix), plotted as a function of the vigor of the saccade (bin size = 0.05 vigor units). Error bars are SEM.

To test our theory further, we thought it useful to have a way to alter decisions in one direction (say work longer) but change movement vigor in the opposite direction (move faster). In theory, this is possible: if the subject is hungry (darker lines in *Figure 2C and D*), that is, the reward is more valuable, then they should again work longer before initiating harvest. Paradoxically, they should also move faster.

In summary, if decisions and actions are coordinated via a policy that aims to maximize the capture rate, then in response to an increased cost of harvest, one should work longer, but move with reduced vigor. In response to an increased reward value, as in hunger, one should also work longer, but now move with increased vigor.

## Increased effort cost of harvest promoted work but reduced saccade vigor

To vary the effort cost of harvest, we altered the tube distance to the mouth (but kept it constant during each session). Varying tube distance affected the decisions of the subjects: when the tube was placed farther, they chose to work longer before starting harvest (*Figure 3A*, left subplot): they attempted more trials during each work period (ANOVA, subject M: $F_{(2,7908)} = 41.5$, $p=5.2 \times 10^{-25}$, subject R: $F_{(2,10948)} = 88.2$, $p=7 \times 10^{-50}$) and produced more successful trials per work period (ANOVA, subject M: $F_{(2,7908)} = 63$, $p=2.8 \times 10^{-24}$, subject R: $F_{(2,10948)} = 163$, $p<10^{-50}$). This policy of delayed gratification was present throughout the recording session (*Figure 3A*, middle plot). That is, when the harvest required more effort, the subjects worked longer to stockpile more food before initiating their harvest (*Figure 3A*, right plot, effect of tube distance on food cached: subject M: $F_{(2,9566)} = 176$, $p<10^{-50}$, subject R: $F_{(2,8907)} = 204$, $p<10^{-50}$).

During the work period, the subjects made saccades to visual targets and accumulated their food. They also made saccades that were not toward visual targets and thus were not eligible for reward. For each animal, we computed the relationship between peak saccade velocity and saccade amplitude across all sessions and then calculated the vigor of each saccade: defined as the ratio of the actual peak velocity with respect to the expected peak velocity for that amplitude (*Reppert et al., 2015*; *Reppert et al., 2018*). For example, a saccade that exhibited a vigor of 1.10 had a peak velocity that was 10% greater than the average peak velocity of the saccades of that amplitude for that subject. As expected, the reward-relevant saccades, that is, saccades made to visual targets (primary, corrective, and center saccades), were more vigorous than other saccades (*Figure 3B*, two-way ANOVA, effect of saccade type, subject M: $F_{(1,391459)} = 7,248$, $p<10^{-50}$, subject R: $F_{(1,355839)} = 13,641$, $p<10^{-50}$).

As a work period began, the reward-relevant saccades exhibited high vigor, but then trial-by-trial, this vigor declined, reaching a low vigor value just before the work period ended (*Figure 3C* vigor). Remarkably, on days in which the tube was placed farther, saccade vigor was lower (RMANOVA, effect of tube distance, subject M: $F_{(2,59033)} = 224$, $p<10^{-50}$, subject R: $F_{(2,50103)} = 75.51$, $p=1.8 \times 10^{-33}$). Thus, increasing the effort cost of extracting food during the harvest period reduced saccade vigor during the work period.

By definition, a more vigorous saccade had a greater peak velocity. This might imply that high vigor saccades should suffer from inaccuracy due to signal dependent noise (*Harris and Wolpert, 1998*). However, we observed the opposite tendency: as saccade vigor increased, both the magnitude and the variance of the endpoint error decreased (*Figure 3C*, two-way ANOVA, effect of vigor on error magnitude, subject M: $F_{(8,59046)} = 480$, $p<10^{-50}$, subject R: $F_{(8,50184)} = 252$, $p<10^{-50}$, effect of vigor on error variance, subject M: $F_{(8,2673)} = 18,200$, $p<10^{-50}$, subject R: $F_{(8,2673)} = 4170$, $p<10^{-50}$). That is, reducing the effort costs of harvest not only promoted vigor, it also facilitated accuracy (*Wang et al., 2016*).

Cognitive signals such as effort and reward are associated with changes in pupil size (*Joshi and Gold, 2020*), as well as transient activation of brainstem neuromodulatory circuits in locus coeruleus (*Bornert and Bouret, 2021*). We wondered if the changes in tube position altered the output of these neuromodulatory circuits, as inferred via pupil size. For each reward-relevant saccade, we measured the pupil size during a ±250 ms window centered on saccade onset, and then normalized this measure based on the distribution of pupil sizes that we had measured during the entire recording for that session in that subject, resulting in a z-score.

At the onset of each work period, the pupils were dilated, but as the subjects performed more trials, the pupils constricted, exhibiting a trial-by-trial reduction that paralleled the changes in saccade

**Figure 4.** As the effort cost of harvest increased, lick vigor declined and the pupils constricted. (**A**) Peak speed of reward-seeking and grooming licks during protraction and retraction as a function of lick amplitude. (**B**) Vigor of reward-seeking licks (protraction) and pupil size as a function of lick number during harvest at various tube distances. (**C**) Lick vigor and pupil size as a function of time during the entire recording session. Line colors depict tube distance as in (**B**). (**D**) Average lick vigor and pupil size during a harvest as a function of number of trials successfully completed in the previous work period. Lick vigor and pupil size were greater when more food had been stored. (**E**) Following a successful lick (contact with food), the next lick was more vigorous and pupils dilated. Following a failed lick, the next lick was slowed and pupils were less dilated. (**F**) We observed no consistent effect of lick vigor on lick accuracy across subjects or across tube distances. Error bars are SEM.

vigor (*Figure 3C*). Notably, the effort cost of harvest affected pupil size: during the work period, the pupils were more dilated if the tube was placed closer to the mouth (*Figure 3C*, RMANOVA, effect of tube distance, subject M: $F(2,60502) = 20$, $p=2 \times 10^{-9}$, subject R: $F(2,50431) = 23.8$, $p=4.9 \times 10^{-11}$). That is, when the effort cost of harvest was lower, the pupils dilated, and the saccades were invigorated.

In summary, when we increased the effort cost of harvest, both the movements and the decisions changed: the pupils constricted and the movements slowed, but they chose to work more trials before initiating harvest.

## Increased effort cost of harvest reduced lick vigor

The work period ended when the subject chose to stop tracking the target and initiated harvest via a licking bout. As in saccades, we defined lick vigor via the ratio of the actual peak velocity of the lick with respect to the expected velocity for that lick amplitude. As amplitude increased, lick peak velocity increased during both protraction

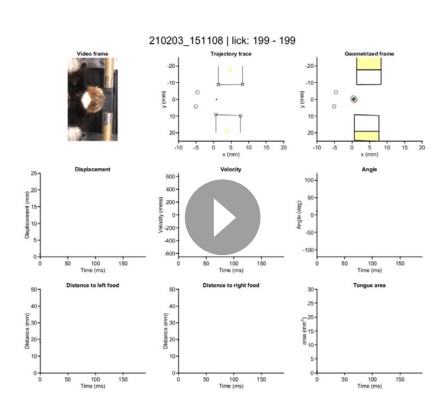

**Animation 5.** Example of a grooming lick.

and retraction (*Figure 4A*). Some of the licks were reward seeking and directed toward the tube, while others were grooming licks, cleaning the tongue and the area around the mouth (*Animation 5*). Reward-seeking licks were more vigorous than grooming licks (two-way ANOVA, effect of lick type, protraction, subject M: $F(1,272233) = 66$, $p=4.5 \times 10^{-16}$, subject R: $F(1,229052) = 698$, $p<10^{-50}$), and retraction was more vigorous than protraction (reward-seeking licks, retraction vs. protraction, subject M: $t(241145) = 532$, $p<10^{-50}$, subject R: $t(213674) = 665$, $p<10^{-50}$).

As the harvest began, the first lick was very low vigor, but lick after lick, the movements gathered velocity, reaching peak vigor by the third or the fourth lick (*Figure 4B*). As the harvest continued, lick vigor gradually declined. Like saccades, licks had a lower vigor in sessions in which the tube was placed farther from the mouth (RMANOVA, effect of tube distance, subject M: $F(2,59033) = 222.5$, $p<10^{-50}$, subject R: $F(2,133502) = 224$, $p<10^{-50}$), and this pattern was present during the entire recording session (*Figure 4C*, left subplot). Thus, an increased effort cost of harvest promoted sloth: reduced vigor of saccades during the work period and reduced vigor of licks during the harvest period.

For each reward-seeking lick, we measured pupil size during a ±250 ms window centered on the moment of peak tongue displacement. During licking, the pupil size changed with a pattern that closely paralleled lick vigor: as the harvest began, pupil size was small, but it rapidly increased during the early licks, then gradually declined as the harvest continued (*Figure 4B*, right subplot). Importantly, the pupils were more dilated in sessions in which the tube was closer to the mouth (*Figure 4C*, right subplot, effect of tube distance, subject M: $F(2,166742) = 583$, $p<10^{-50}$, subject R: $F(2,130493) = 118$, $p<10^{-50}$). As a result, when the effort cost of reward increased, the pupils constricted, and the vigor of both saccades and licks decreased.

While the theory predicted that moving the tube farther would result in a longer work period and reduced movement vigor, it also predicted that the subjects would reduce their harvest duration (reduced licks, *Figure 2C*). That is, it predicted that the subjects would work longer, stowing more food, but leave more of it behind. This last prediction did not agree with our data (see 'Discussion'). For subject R, the number of licks was approximately the same across the various tube distances, and for subject M the number of licks increased with tube distance (*Figure 1—figure supplement 1*).

In summary, within a harvest period, lick vigor rapidly increased and then gradually declined. Simultaneous with the changes in vigor, the pupils rapidly dilated and then gradually constricted. In sessions where the tube was placed farther from the mouth, the licks had lower vigor and the pupils were more constricted.

## Expectation of greater reward increased lick vigor

As the subject worked, they accumulated food, thus increasing the magnitude of the available reward. To check whether reward magnitude affected movement vigor, for each tube distance we computed the average lick vigor during the harvest as a function of the number of trials completed in the preceding work period. We found that when the work period had included many completed trials, then the movements in the ensuing harvest period were more vigorous (*Figure 4D*, two-way ANOVA, effect of trials, subject M: $F(4, 164242) = 353$, $p<10^{-50}$, subject R: $F(4,123411) = 152$, $p<10^{-50}$). Thus, the licks were invigorated by the amount of food that awaited harvest.

Because the tube was small, many of the licks missed their goal and failed to contact the food. The success or failure of a lick affected both the vigor of the subsequent lick and the change in the size of the pupil. Following a successful lick, there was a large increase in lick vigor (*Figure 4E*, subject M: $t(85182) = 40$, $p<10^{-50}$, subject R: $t(81378) = 104$, $p<10^{-50}$), and a large increase in pupil size (subject M: $F(84969) = 57$, $p<10^{-50}$, subject R: $t(80318) = 94$, $p<10^{-50}$). In contrast, following a failed lick the subjects either reduced or did not increase their lick vigor (*Figure 4E*, subject M: $t(114159) = 0.88$, $p=0.37$, subject R: $t(97164) = -44$, $p<10^{-50}$). This failure also produced a smaller increase in pupil size (comparison to successful lick, two-sample *t*-test, subject M: $t(198722) = 14.8$, $p=4.3 \times 10^{-50}$, subject R: $t(176044) = 53$, $p<10^{-50}$). Thus, a single successful lick led to acquisition of reward, which then was followed by a relatively large increase in pupil size, and an invigorated subsequent lick.

For saccades, we had found that increased vigor was associated with greater accuracy. To quantify the relationship between lick vigor and accuracy, for each tube distance we labeled each reward-seeking lick as being high or low vigor. For subject M, high vigor licks tended to be more successful,

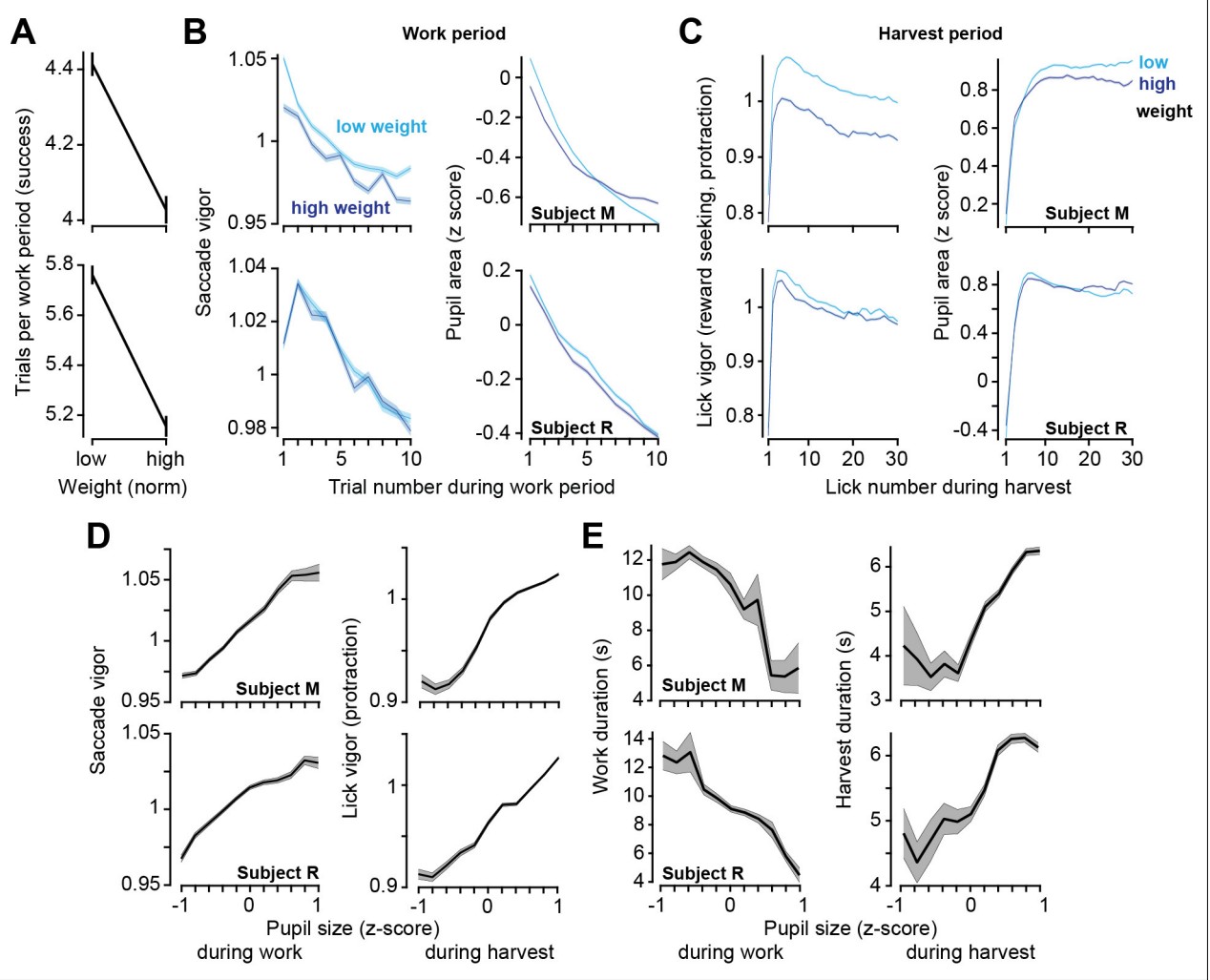

**Figure 5.** Relatively low body weight, potentially reflecting a greater valuation of reward, coincided with longer work periods and greater vigor. Pupil size correlated with both vigor and decisions. (**A**) Trials successfully completed during a work period as a function of normalized body weight at the start of the session. (**B**) Left: saccade vigor as a function of trial number for low and high body weights. Right: pupil size during the same saccades. (**C**) Left: lick vigor as a function of lick number during the harvest period. Right: pupil size during the same licks. (**D**) Saccade vigor during the work period, and lick vigor during the harvest period, as a function of pupil size. (**E**) Work duration and harvest duration as a function of pupil size. Error bars are SEM.

but this was not the case for subject R (*Figure 4F*). Moreover, tube distance did not produce a consistent effect on lick success.

In summary, the subject licked more vigorously following a long work period in which they had accumulated more reward. Moreover, when a lick was successful in acquiring reward, they increased the vigor of the subsequent lick.

## Hunger promoted work and increased vigor

Our theory predicted that it should be possible to change decisions in one direction (say work longer), while altering movement vigor in the opposite direction (move faster). An increase in the subjective value of reward, as might occur when the subject is hungry, should have two effects: increase the number of trials that the subject chooses to perform before commencing harvest and increase movement vigor.

We did not explicitly manipulate the weight of the subjects. Indeed, to maintain their health, we strived to keep their weights constant during the roughly 2.5-year period of these experiments. However, there was natural variability, which allowed us to test the predictions of the theory.

We found that when their weight was lower than average, the subjects chose to work a greater number of trials before commencing harvest (*Figure 5A*, two-sample *t*-test, subject M: t(11052) = 7.9, p=3.4 × 10$^{-25}$, subject M: t(12549) = 10.1, p=9.3 × 10$^{-24}$). This result was similar to the effect that we had seen when the effort cost of harvest was increased. However, the theory had predicted that the effect on vigor should be in the opposite direction: if hunger increased reward valuation, then one should speed the movements and hasten food acquisition. Notably, weight did not have a consistent effect on saccade vigor across the two subjects (*Figure 5A*), yet during the harvest, both subjects licked with greater vigor when their weight was lower (*Figure 5A*, subject M: t(219752) = 88, p<10$^{-50}$, subject R: t(205163) = 22, p<10$^{-50}$).

Thus, while both the effort cost of reward and hunger promoted greater work, effort promoted sloth while hunger promoted lick vigor.

## Pupil size variations strongly correlated with changes in decisions and movements

Finally, we considered the data across both the work and the harvest periods and asked how well movement vigor tracked pupil size. The results demonstrated that in both the work and the harvest periods, for both saccades and licks, an increase in pupil size was associated with an increase in vigor (*Figure 5B*, reward-relevant saccades, subject M: *r* = 0.989, p=7.7 × 10$^{-9}$, subject R: *r* = 0.97, p=7.1 × 10$^{-7}$; reward-seeking protraction licks, subject M: *r* = 0.969, p=9.8 × 10$^{-7}$, subject R: *r* = 0.989, p=6.3 × 10$^{-9}$). Moreover, when the pupil was dilated, the work periods tended to be shorter (*Figure 5C*, subject M: *r* = −0.90, p=0.00014, subject R: *r* = −0.97, p=6.3 × 10$^{-7}$), while harvest durations tended to be longer (*Figure 5C*, subject M: *r* = 0.894, p=0.00021, subject R: *r* = 0.935, p=2.4 × 10$^{-5}$). Thus, pupil dilation was associated with choosing to work less, while moving faster.

## Discussion

What we choose to do is the purview of the decision-making circuits of our brain, while the implicit vigor with which we perform that action is the concern of the motor-control circuits. From a theoretical perspective (*Yoon et al., 2018*), our brain should coordinate these two forms of behavior because both the act that we select and its vigor dictate expenditure of time and energy, contributing to a capture rate that affects longevity and fecundity (*Lemon, 1991*). Does the brain coordinate decisions and movements to maximize a capture rate? If so, how might the brain accomplish this coordination?

Here, we designed a foraging task in which marmosets worked by making saccades, accumulating food for each successful trial, then stopped working and harvested their cache by licking. On every trial they decided whether to work or to harvest. Their decision was carried out by the motor system, producing either a visually guided saccade or a lick, each exhibiting a particular vigor. The theory predicted that to maximize the capture rate, the appropriate response to an increased effort cost of harvest was to do two things: work longer to cache more food but reduce vigor to conserve energy.

We varied the effort costs by moving the food tube with respect to the mouth. This changed the effort cost of harvest but not the effort cost of work. The subjects responded by altering how they worked as well as how they harvested. When the harvest was more effortful, they performed more saccade trials to stockpile food. They also slowed their movements, reducing saccade velocity during the work period, and reducing lick velocity during the harvest period. Notably, the most vigorous saccades were also the most accurate: as saccade vigor increased, so did endpoint accuracy.

The theory made a second prediction: as the value of reward increased, the subjects should again choose to work a longer period before initiating harvest, but unlike the effort costs, now respond by moving more vigorously. We did not directly manipulate the subjective value of reward, but rather relied on the natural fluctuations in body weight and assumed that when their weight was low, the subjects were hungrier for reward. Indeed, when their weight was low, the subjects again chose to work longer, but now elevated their vigor during the harvest period.

Finally, we quantified the effect of reward magnitude on vigor. Within a session, lick vigor increased robustly as a function of the number of trials completed in the preceding work period. Thus, the licks were invigorated by the amount of food that awaited harvest.

Notably, some of the predictions of the theory did not agree with the experimental data. An increased effort cost did not accompany a reduction in the duration of harvest, and hunger did not

increase saccade vigor robustly. Indeed, earlier experiments have shown that if the effort cost of harvest increases, animals who expend the effort will then linger longer to harvest more of the reward that they have earned (*Cowie, 1977*). This mismatch between observed behavior and theory highlights some of the limitations of our formulation. For example, our capture rate reflected a single work-harvest period rather than a long sequence. Moreover, the capture rate did not consider the fact that the food tube had finite capacity, beyond which the food would fall and be wasted. This constraint would discourage a policy of working more but harvesting less. Finally, if we assume that a reduced body weight is a proxy for increased subjective value of reward, it is notable that we observed a robust effect on vigor of licks, but not saccades. A more realistic capture rate formulation awaits simulations, possibly one that describes capture rate not as the ratio of two sums (sum of gains and losses with respect to sum of time), but rather the expected value of the ratio of each gain and loss with respect to time (*Bateson and Kacelnik, 1995*; *Bateson and Kacelnik, 1996*).

A shortcoming of our model is that we did not include a link between lick vigor and its probability of success. As a result, when we moved the food tube away, the model did not consider the possibility that maintaining lick accuracy may involve reduced vigor. The reason for this is that we searched for but could not find a consistent relationship, across subjects or effort conditions, between protraction speed of the tongue and its success probability. Thus, we cannot exclude this alternate hypothesis. However, the most interesting aspect of our results was that when we increased tube distance, making harvest more effortful, there was not only a reduction in lick vigor, but also a reduction in saccade vigor. That is, the decisions and actions during the work period responded to the increased effort cost of reward during the harvest period.

What might be a neural basis for this coordination of decisions and movements? A clue was the fact that the pupils were more constricted in sessions in which the effort cost of harvest was greater. This global change in pupil size accompanied delayed harvest and reduced vigor across sessions, but surprisingly, even within a session, transient changes in pupil size accompanied changes in vigor. During the work period, the trial-to-trial reduction in saccade vigor accompanied trial-to-trial constriction of the pupil, and within a harvest period, the rapid rise and then the gradual fall in lick vigor paralleled rapid dilatation followed by gradual constriction of the pupil.

Pupil dilation is a proxy for activity in the brainstem neuromodulatory system (*Vazey et al., 2018*) and is a measure of arousal (*Mathôt, 2018*). Control of pupil size is dependent on spiking of norepinephrine neurons in locus coeruleus (LC-NE): an increase in the activity of these neurons produces pupil dilation (*Joshi et al., 2016*; *Breton-Provencher and Sur, 2019*). Some of these neurons show a transient change in their activity when acquisition of reward requires expenditure of either physical (*Bornert and Bouret, 2021*) or mental effort (*Contadini-Wright et al., 2023*), even when there is no concomitant movement to be made. It is possible that in the present task, as the effort cost of harvest increased, LC-NE neurons decreased their activity, producing pupil constriction. If so, the reduced NE release may have had two simultaneous effects: encourage work and promote delayed gratification in brain regions that control decisions, discourage energy expenditure and promote sloth in brain regions that control movements. Thus, the idea that emerges is that the response of NE to economic variables, as inferred via changes in pupil size, might act as a bridge to coordinate the computations in the decision-making circuits with the computations in the motor-control circuits, aiming to implement a consistent control policy that improves the capture rate.

In addition to NE, the basal ganglia and, in particular, the neurotransmitter dopamine are likely the key contributors to the coordination of decisions with actions (*Thura and Cisek, 2017*; *Herz et al., 2022*). When the effort price of a preferred food increases, animals choose to work longer, pressing a lever a greater number of times (*Salamone et al., 1991*; *Aberman and Salamone, 1999*). This desire to expend effort to acquire a valuable reward is reduced if dopamine is blocked in the ventral striatum (*Koch et al., 2000*; *Farrar et al., 2010*; *Yohn et al., 2015*). Hunger activates circuits in the hypothalamic nuclei, disinhibiting dopamine release in response to food cues (*Cassidy and Tong, 2017*). Dopamine concentrations in the striatum drop when the effort price of a food reward increases (*Schelp et al., 2017*), and dopamine release before onset of a movement tends to invigorate that movement (*da Silva et al., 2018*). Thus, the presence of dopamine may not only alter decisions by encouraging expenditure of effort, but also modify movements by promoting vigor.

Experiments of *Hayden et al., 2011* and *Barack et al., 2017* suggest that the decision of when to stop work and commence harvest may rely on computations that are carried out in the cingulate

cortex. They found that as monkeys deliberated between the choice of staying and acquiring diminishing rewards, or leaving and incurring a travel cost, these neurons encoded a decision variable that reflected the value of leaving the patch. The prediction that emerges from our work is that the rate of rise of these decision variables may be modulated by the presence of NE.

From a motor-control perspective, a surprising aspect of our results was that an increase in saccade vigor accompanied an improvement in endpoint accuracy. In our earlier work, we found that during reaching, reward increased vigor without reducing accuracy (*Summerside et al., 2018*). Thus, the brain has the means to increase movement vigor and improve its accuracy. How is this achieved?

We found that the high vigor saccades were produced when the pupils were dilated, implying an increased release of NE. In songbirds, increased NE release acts on the basal ganglia to suppress activity of spiny neurons, and this reduced activity in the basal ganglia accompanies reduced variance in the songs that the animal sings (*Singh Alvarado et al., 2021*). Thus, NE may play a critical role in control of movement variability. For saccades, control of endpoint accuracy depends on the coordinated activity of Purkinje cells in the oculomotor region of the cerebellar vermis (*Sedaghat-Nejad et al., 2022*; *Barash et al., 1999*). LC projects to the cerebellum, and stimulation of LC neurons increases the sensitivity of Purkinje cells to their inputs (*Moises et al., 1981*).

Is movement vigor increased following increased NE inputs from LC to the basal ganglia, and accuracy improved following increased NE inputs from LC to the cerebellum? Does decision-making shift toward greater work and delayed gratification following reduced NE inputs from LC to the frontal lobe? These are some of the questions that await future experiments.

## Methods

Behavioral and neurophysiological data were collected from two marmosets (*Callithrix jacchus*, male and female, 350–390 g, subjects R and M, 6 years old). The neurophysiological data focused on the cerebellum and are described elsewhere (*Sedaghat-Nejad et al., 2022*; *Sedaghat-Nejad et al., 2019*; *Muller et al., 2023*). Here, our focus is on the behavioral data.

The marmosets were born and raised in a colony that Prof. Xiaoqin Wang has maintained at the Johns Hopkins School of Medicine since 1996. The procedures on the marmosets were evaluated and approved by the Johns Hopkins University Animal Care and Use Committee, protocol number PR22M285, in compliance with the guidelines of the United States National Institutes of Health.

### Data acquisition

Following recovery from head-post implantation surgery, the animals were trained to make saccades to visual targets and rewarded with a mixture of apple sauce and lab diet (*Sedaghat-Nejad et al., 2019*). They were placed in a monkey chair and head-fixed while we presented visual targets on an LCD screen (Curved MSI 32″ 144 Hz, model AG32CQ) and tracked both eyes at 1000 Hz using an EyeLink-1000 system (SR Research, USA). The timing of target presentation on the video screen was measured using a photo diode. Tongue movements were tracked with a 522 frame per second Sony IMX287 FLIR camera, with frames captured at 100 Hz.

Each trial began with a saccade to the center target followed by fixation for 200 ms, after which a primary target (0.5 × 0.5° square) appeared at one of eight randomly selected directions at a distance of 5–6.5°. Onset of the primary target coincided with the presentation of a tone. As the animal made a saccade to the primary target, that target was erased and a secondary target was presented at a distance of 2–2.5°, also at one of eight randomly selected directions. The subject was rewarded if following the primary saccade it made a corrective saccade to the secondary target, landed within 1.5° radius of the target center, and maintained fixation for at least 200 ms. Onset of reward coincided with the presentation of another distinct tone. Following an additional 150–250 ms period (uniform random distribution), the secondary target was erased and the center target was displayed, indicating the onset of the next trial. Thus, a successful trial comprised of a sequence of three saccades: center, primary, and corrective, after which the subject received a small increment of food (0.015 mL).

The food was provided in two small tubes (4.4 mm diameter), one to the left and the other to the right of the animal (*Figure 1A*). A successful trial produced a food increment in one of the tubes and would continue to do so for 50–300 consecutive trials, then switch to the other tube. Because the food increment was small, the subjects naturally chose to work for a few consecutive trials, tracking the

visual targets and allowing the food to accumulate, then stopped tracking and harvested the food via a licking bout. The licking bout typically included a sequence of 15–40 licks. The subjects did not work while harvesting. As a result, the behavior consisted of a work period (targeted saccades), followed by a harvest period (targeted licking), repeated hundreds of times per session.

The critical variables were the number of trials that the subject chose to perform before initiating harvest, the vigor of their saccades during the work period, and the vigor of their licks during the harvest period.

## Data analysis

All saccades, regardless of whether they were instructed by presentation of a visual target or not, were identified using a velocity threshold. Saccades to primary, secondary, and central targets were labeled as reward-relevant saccades, while all remaining saccades were labeled as task irrelevant.

We analyzed tongue movements using DeepLabCut (*Mathis et al., 2018*). Our network was trained on 89 video recordings of the subjects with 15–25 frames extracted and labeled from each recording. The network was built on the ResNet-152 pre-trained model, and then trained over $1.03 \times 10^6$ iterations with a batch size of 8, using a GeForce GTX 1080Ti graphics processing unit (*He et al., 2016*). A Kalman filter was further applied to improve quality and smoothness of the tracking, and the output was analyzed in MATLAB to quantify varying lick events and kinematics.

We tracked the tongue tip and the edge of the food in the tube, along with control locations (nose position and tube edges). We tracked all licks, regardless of whether they were aimed toward the tube (reward seeking) or not (grooming). Reward-seeking licks were further differentiated based on whether they aimed to enter the tube (inner-tube licks), hit the outer edge of the tube (outer-edge licks), or fell below the tube (under tube). If any of these licks successfully contacted the tube, we labeled that lick as a success (otherwise, a failed lick).

Pupil area was measured during a ±250 ms period centered at the onset of each reward-relevant saccade and the onset of each lick. We then normalized the pupil measurements by representing it as a z-score with respect to the mean value for that session.

## Saccade and tongue vigor

We relied on previous work to define vigor of a movement (*Yoon et al., 2020*; *Yoon et al., 2018*; *Reppert et al., 2015*; *Reppert et al., 2018*). Briefly, if the amplitude of a movement is $x$ and the peak speed of that movement is $v$, then for each subject the relationship between the two variables can be described as:

$$v = \alpha \left( 1 - \frac{1}{1 + \beta x} \right) \tag{3}$$

In the above expression, $\alpha, \beta \geq 0$ and are subject-specific parameters. For a movement with amplitude $x$, its vigor was defined as the ratio of the actual peak speed with respect to the expected value of its peak speed, that is, $\frac{v}{E[v(x)]}$. Expected value was computed by fitting *Equation 3* to all the data acquired across all sessions. When vigor is greater than 1, the movement had a peak velocity that was higher than the mean value associated with that amplitude.

## Model formulation

We chose a formulation of utility (*Equation 1*) based on a normative approach that ecologists have used to understand the decisions that animals make regarding how far to travel for food, what mode of travel to use, and how long to stay before moving on to another reward opportunity (*Richardson and Verbeek, 1986*; *Stephens and Krebs, 1987*; *Bautista et al., 2001*). In a typical formulation of the theory, the numerator represents the reward gained (in units of energy), minus the effort expended (also in units of energy), while the denominator represents the amount of time spent during that behavior. We represented this idea in *Equation 1* with saccades that produced reward accumulation and licks that produced reward consumption. Thus, the utility that we aim to maximize is the rate of energy gained.

The specific functions that we used to represent the energy gained through reward acquisition and the energy expended through effort expenditure came either from experiment design or from the measurements we have made in other experiments. We modeled reward accumulation as a linear

rise in energy stored because successful saccades produced a linear increase in the food cache. We modeled harvesting of the food as a hyperbolic function of the number of licks to represent the fact that as the licking bout began, each successful lick depleted the food, and thus the first few licks produced a greater amount of food consumption than the last few licks. We modeled the effort cost of licking as a linear function of the number of licks.

A critical assumption that we made is that energy expended performing the saccade trials (which grew faster than linearly as a function of the number of trials attempted) grew faster than the time spent attempting those same trials (which grew linearly with the number of trials). This assumption is based on the heuristic that the average rate of energy lost following a large number of attempted trials is greater than the average rate of energy lost following a small number of attempted trials.

The model's simplicity provided closed-form solutions across all parameter values, allowing us to make predictions without having to fit the model to the measured data. For example, for all parameter values that produce a real solution (as opposed to imaginary), the optimal number of saccade trials increases with the square root of the cost of licking. Thus, the basic prediction of the model is that to maximize the capture rate, regardless of parameter values, an increase in the effort required for harvest should be met with a greater willingness to work. The closed-form solutions are presented in the supplementary document (simulations.nb).

## Other models of utility

In composing our utility (*Equation 1*), we chose to combine reward and effort additively. This is in contrast to other approaches in which effort discounts reward multiplicatively (*Sugiwaka and Okouchi, 2004*; *Prévost et al., 2010*; *Klein-Flügge et al., 2015*). Our reasoning is that multiplicative interactions have the limitation that they are incompatible with the observation that reward invigorates movements.

To compare additive and multiplicative approaches, let us consider an arbitrary function $U(T)$ that specifies how effort varies with movement duration $T$. Typically, this is a U-shaped function that describes energy expenditure as a function of movement duration, as in *Shadmehr et al., 2016*. In the case of multiplicative interaction between reward and effort, we can consider the following representation of utility:

$$J = \frac{\alpha}{1+T} U^{-1}(T) \tag{4}$$

In the above formulation, reward $\alpha$ is discounted hyperbolically with time and an increase in reward increases the utility of the action. The optimum movement vigor has the duration $T^*$ that maximizes this utility. Notably, because increasing reward merely scales this utility, it has no effect on vigor. Thus, a utility in which reward is multiplied by a function of effort generally fails to predict dependence of movement vigor on reward.

## Simulations

The optimal policy specifies the decisions and movements that for the effort cost defined in *Equation 2* maximizes the capture rate defined in *Equation 1*. This policy selects the number of saccade trials $n_s$ to perform during the work period, the number of licks $n_l$ to perform during the harvest period, and the vigor of each lick, represented by the average duration of a lick $T_l$. To compute the optimal policy, we found the derivative of the capture rate with respect to each policy variable $n_s$, $n_l$, and $T_l$, then set each derivative equal to zero, producing three simultaneous nonlinear equations. In all three cases, we were able to solve for the relevant control variable analytically (see *Supplementary file 1* for the derivations). We found that if the solution was a real number, then regardless of parameter values, an increase in $d$ (distance of the tube to the mouth), the optimal policy produced an increase in $n_s^*$, decrease in $n_L^*$, and increase in $T_L^*$. Thus, the results illustrated in *Figure 2* are robust to changes in parameter values.

To generate the plots in *Figure 2A*, we used the following parameter values: $\alpha = 20$, $\beta_s = 0.5$, $\beta_L = 0.3$, $c_s = 0.5$, $T_s = 1$, $T_L = 0.2$, $c_L = 0.5$ (low effort), and $c_L = 2.5$ (high effort). For the plots in *Figure 2C and D*, we used the same parameter values, but $c_L$ was defined via *Equation 2*. Thus, tube distance $d$ varied, and $T_L$ was unknown and was solved for. In *Equation 2*, $k_L = 1$. In the simulations, to describe state of hunger, we set $\alpha = 20$ for a sated state and $\alpha = 25$ for a hungry state.

## Statistical analysis

Hypothesis testing was performed using the functions provided by the *MATLAB Statistics and Machine Learning Toolbox,* version R2021b. For *t*-tests, across the one-sample, paired-sample, and two-sample conditions, p-values were computed using the *ttest* and *ttest2* functions with data that was combined across sessions, separated by condition. For ANOVA, in the one-way condition, p-values were computed using a nonparametric Kruskal–Wallis test, using the *kruskalwallis* function. In the two-way condition, the *anovan* function was used to compute p-values, accounting for an unbalanced design resulting from a varied number of samples across conditions. In both cases, like in the *t*-tests, data was combined across sessions, separated by condition. In the repeated measures condition, each session was treated as a subject with multiple repeated measures representing a given variable (i.e., lick vigor per lick in a harvest period). To fit a repeated measures model, the *fitrm* function was used, then analyzed using the *ranova* function. In all cases of repeated measures ANOVA, compound symmetry assumptions were tested using the Mauchly sphericity test with the *maulchy* function. In cases where the assumption was violated (Maulchy test $p<0.05$), epsilon adjustments were used, with the *epsilon* function, to compute corrected p-values (for $\varepsilon > 0.75$, use Huynh–Feldt p-value; and for $\varepsilon < 0.75$, use Greenhouse–Geisser p-values). For correlation analyses, Pearson's correlation coefficient, $r$, and corresponding p-values were computed using the *corrcoef* function.

## Acknowledgements

The work was supported by grants from the NIH (R01-EB028156, R01-NS078311, R37-NS128416) and the Office of Naval Research (N00014-15-1-2312).

## Additional information

### Funding

| Funder | Grant reference number | Author |
|---|---|---|
| National Institutes of Health | R01-EB028156 | In Kyu Jang<br>Reza Shadmehr<br>Paul Hage<br>Vivian Looi<br>Jay S Pi<br>Mohammad Amin Fakharian |
| National Institutes of Health | R01-NS078311 | Reza Shadmehr<br>Paul Hage<br>Mohammad Amin Fakharian<br>Jay S Pi |
| National Institutes of Health | R37-NS128416 | Paul Hage<br>In Kyu Jang<br>Vivian Looi<br>Mohammad Amin Fakharian<br>Jay S Pi<br>Reza Shadmehr |
| Office of Naval Research | N00014-15-1-2312 | Simon P Orozco<br>Ehsan Sedaghat-Nejad<br>Reza Shadmehr |

The funders had no role in study design, data collection and interpretation, or the decision to submit the work for publication.

### Author contributions

Paul Hage, Conceptualization, Data curation, Software, Formal analysis, Writing - review and editing; In Kyu Jang, Data curation, Software; Vivian Looi, Jay S Pi, Software, Formal analysis; Mohammad Amin Fakharian, Data curation, Software, Formal analysis; Simon P Orozco, Ehsan Sedaghat-Nejad,

Data curation; Reza Shadmehr, Conceptualization, Funding acquisition, Writing - original draft, Writing - review and editing

### Author ORCIDs
Reza Shadmehr ⬡ http://orcid.org/0000-0002-7686-2569

### Ethics
The procedures on the marmosets were evaluated and approved by the Johns Hopkins University Animal Care and Use Committee in compliance with the guidelines of the United States National Institutes of Health. protocol number PR22M285.

Reviewer #1 (Public Review): https://doi.org/10.7554/eLife.87238.3.sa1
Reviewer #2 (Public Review): https://doi.org/10.7554/eLife.87238.3.sa2
Author Response https://doi.org/10.7554/eLife.87238.3.sa3

## Additional files

### Supplementary files
• Supplementary file 1. Mathematica notebook simulations for optimal foraging.
• MDAR checklist

### Data availability
Data are available at Open Science Framework: https://osf.io/54JS6.

The following dataset was generated:

| Author(s) | Year | Dataset title | Dataset URL | Database and Identifier |
| --- | --- | --- | --- | --- |
| Hage P, Shadmehr R | 2023 | Effort cost of harvest affects decisions and movement vigor of marmosets during foraging | https://osf.io/54JS6/ | Open Science Framework, 54JS6 |

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
