## [Editor Report · eLife assessment]

This **important** study unravels the interaction between effort cost, pupil-indexed brain state, and movement (saccadic) vigor during foraging decisions in marmoset monkeys. Based on a normative computational model, the authors derive the prediction that anticipated effort should affect both decisions and movement vigor during foraging; and then provide **solid** behavioral and pupillometric evidence for this prediction in a foraging task. This paper will be of interest to decision and motor neuroscience as well as to all researchers studying animal behavior.

---

## [Referee Report · Reviewer #1 (Public Review)]

The manuscript by Hage et al. presents interesting results from a foraging behavior in Marmosets that explores the interactions of saccade and lick vigor with pupil dilation and performance as well as a marginal value theory and foraging theory-inspired value-based decision-making model thereof. The results are generally robust and carefully presented and analyses, particularly of vigor, are carefully executed.

---

## [Referee Report · Reviewer #2 (Public Review)]

Hage et al examine how the foraging behavior of marmoset monkeys in a laboratory setting systematically takes into account the reward value and anticipated effort cost associated with the acquisition and consumption of food. In an interesting comprehensive framework, the authors study how experimental and natural variation of these factors affect both the decisions and actions necessary to gather and accumulate food, as well as the actions necessary to consume the food.

The manuscript proposes a computational model of how the monkeys may guide all these aspects of behavior, by maximizing a food capture rate that trades off the food that can be gathered with the effort and duration of the underlying actions. They use this model to derive qualitative predictions for how monkeys should react to an increase in the effort associated with food consumption: Monkeys should work longer before deciding to consume the accumulated food, but should move more slowly. The model also predicts that monkeys should show a different reaction to an increase in reward value of the food, also working longer but moving faster. The authors test these predictions in an interesting experimental setup that requires monkeys to collect small increments of food rewards for successful eye movements to targets. The monkeys can decide freely when to interrupt work and consume the accumulated food, and the authors measure the speed of the eye movements involved in the food acquisition as well as the tongue movements involved in the food consumption.

By and large, the behavioral findings fall in line with the qualitative model predictions: When the effort involved in food consumption increases, monkeys collect more food before deciding to consume it, and they move slower both during food acquisition and food consumption. In a second test, the authors approximate the effects of reward value of the food at stake, by comparing monkey behavior during different days with natural variations in body weight. These quasi-experimental increases in the reward value of food also lead to longer work times before consumption, but to faster movements during food consumption. Finally, the authors show that these effects correlate with pupil size, with pupils dilating more for low-effort foraging actions with increased saccade speed and decreased work duration. The authors conclude that the effort associated with anticipated actions can lead to changes in global brain state that simultaneously affect decisions and action vigor.

The paper proposes an interesting model for how one unified action policy may simultaneously affect multiple types of decisions and movements involved in foraging. The methods employed to measure behavior and test these predictions are generally sound, and the paper is well written.

---

## [Author Response]

The following is the authors’ response to the original reviews.

**Reviewer #1 (Public Review):**
The manuscript by Hage et al. presents interesting results from a foraging behavior in Marmosets that explores the interactions of saccade and lick vigor with pupil dilation and performance as well as a marginal value theory and foraging theory-inspired value-based decision-making model thereof. The results are generally robust and carefully presented and analyses, particularly of vigor, are carefully executed.The authors constructed a model that makes two predictions: "In summary, this simple theory made two sets of predictions: in response to an increased cost of harvest, one should work longer, but move with reduced vigor. In response to an increased reward value, as in hunger, one should also work longer, but now move with increased vigor." Their behavioral data meets these predictions. It is not clear if the model was designed and tweaked in order to make those predictions and match the data, or derived from principles. Furthermore, it is not clear what other models would make similar predictions. It would help to assess what is predicted by other simple models, as well as different functional forms for the effort costs in their model.

We chose this formulation of utility (Eq. 1) because it is a normative approach that ecologists have used to understand the decisions that animals make regarding how far to travel for food, what mode of travel to use, and how long to stay before moving on to another reward opportunity (Richardson and Verbeek 1986; Stephens and Krebs 1986; Bautista et al. 2001). In a typical formulation of the theory, the numerator represents the reward gained (in units of energy), minus the effort expended (also in units of energy). The denominator represents the amount of time spent during that behavior. We represented this idea in Eq. (1) with saccades that produced reward accumulation, and licks that produced reward consumption. Thus, the utility that we are trying to maximize is the rate of energy gained.

The specific functions that we used to represent the energy acquired through reward acquisition, and the energy expended through effort expenditure, came a priori either from experiment design, or from the measurements we have made in other experiments. We modeled reward accumulation as a linear rise in energy stored because successful saccades produced a linear increase in the food cache. We modeled consumption of the food as a hyperbolic function of the number of licks to represent the fact that as the licking bout began, each successful lick depleted the food, and thus the first few licks produced a greater amount of food consumption than the last few licks. We modeled the effort cost of licking to grow linearly with the number of licks.

A critical assumption that we made is that energy spent performing the saccade trials (which grew faster than linearly as a function of the number of trials attempted), grew faster than the time spent attempting those same trials (which grew linearly with the number of trials). This assumption is based on the heuristic that the average rate of energy lost following a large number of attempted trials is greater than the average rate of energy lost following a small number of attempted trials.

Sensitivity to parameter values: The model’s simplicity provides closed-form solutions across all parameter values, allowing one to make predictions without having to fit the model to the measured data. For example, for all parameter values that produce a real solution (as opposed to imaginary), the optimal number of saccade trials increases with the square root of the cost of licking. Thus, the basic prediction of the model is that in order to maximize the capture rate, an increase in the effort that it takes to harvest the reward should produce a greater willingness to work longer, caching more food. The closed-form solutions are presented in the Mathematica supplementary document.

Other models of utility: In composing our utility (Eq. 1), we chose to combine reward and effort additively. This is in contrast to other approaches in which effort discounts reward multiplicatively (47–49). Here, let us show that multiplicative interactions may have the limitation that they are incompatible with the observation that reward invigorates movements.To compare additive and multiplicative approaches, let us consider an arbitrary function U(T) that specifies how effort varies with movement duration. Typically, this is a U-shaped function that describes energy expenditure as a function of movement duration, as in Shadmehr et al. (2016). In the case of multiplicative interaction between reward and effort, we can consider the following representation of utility:J=α1+τU−1(T)

In the above formulation, reward α is discounted hyperbolically with time, and an increase in reward increases the utility of the action. The optimum movement vigor has the duration T∗ that maximizes this utility. Notably, because increasing reward merely scales this utility, it has no effect on vigor. Thus, a utility in which reward is multiplied by a function of effort generally fails to predict dependence of movement vigor on reward.

Line 37 page 6; the link of pupil to NE/LC is tenuous. Other modulators systems and circuits may be equally important and should be mentioned (e.g. Reimer, Jacob, Matthew J. McGinley, Yang Liu, Charles Rodenkirch, Qi Wang, David A. McCormick, and Andreas S. Tolias. "Pupil fluctuations track rapid changes in adrenergic and cholinergic activity in cortex." Nature communications 7, no. 1 (2016): 13289.)

Reimer et al. (2016) used two-photon microscopy to measure activity of ACh and NE projections in layer 1 of mouse visual cortex while tracking pupil diameter fluctuations. During stillness, elevated pupil diameter was followed by cholinergic and noradrenergic axonal activity. Notably, NE activity levels were larger and with shorter latency than ACh. In primates, Joshi et al., (2016) recorded from LC during a fixation task. Using spike-triggered averaging, they found that following a spike in an LC neuron, there was pupil dilation at 200-300 ms latency. Moreover, microstimulation in LC produced pupil dilation at 500ms latency. More recently, Breton-Provencher and Sur (2019) provided causal evidence that LC activity drives pupil size. They optogenetically activated (1s) or silenced (5 sec) locus coeruleus noradrenergic neurons and found strong increase in pupil size or modest decrease: increase had a slow time scale of 1 second or more, similar slow timescale for decrease. The LC-NA neurons are surrounded by GABA-ergic neurons. Stimulation of the GABA-ergic neurons produced mild, slow constriction. They identified GABA-ergic and NA neurons by photo-tagging and then tried to identify them via spike shape and found that “spike shape of some GABA neurons were not well separated from NA neurons, demonstrating the difficulty of cell-type identification based on spike shape alone.” They noted that a subset of GABAergic neurons received coincident inputs with the NA neurons. When the GABA neurons were excited, the gain of the pupil response to an auditory tone was diminished, producing an increase as a function of tone intensity that had a lower gain. Thus, LC-NA neurons causally drive pupil size, and the GABA neurons that surround them control the gain of the response of LC-NA neurons to arousal stimuli.

Line 35 page 6-page 7 line 10 emphasizes a cognitive interpretation of the pupil dilations that is emphasized, in relation to effort costs. But there are also more concomitant vigorous movements. Could all of their pupil results be explained by motor correlates? This should be tested and ruled out before making cognitive interpretations.

Pupil dilation is a proxy for activity in the brainstem neuromodulatory system (Vazey et al., 2018) and is a measure of arousal (Mathot, 2018). Control of pupil size is dependent on spiking of norepinephrine neurons in locus coeruleus (LC-NE): an increase in the activity of these neurons produces pupil dilation (Joshi et al., 2016; Breton-Provencher and Sur, 2019). Some of these neurons show a transient change in their activity when acquisition of reward requires expenditure of physical effort (Bornert and Bouret, 2021). However, the link between effort costs and pupil size appears to go beyond motor control, as a recent paper found that pupil size increases during effortful speech perception (Contadini-Wright et al., 2023). Thus, although in our work increases in pupil size were always associated with increased movement vigor, the results from other studies suggest that economic variables such as cognitive effort in tasks in which there is no concomitant movement also drive an increase in pupil size.

Page 7, line 37-42: How would the model need to be modified in order to account for this discrepancy with the data? Ideally, this would be tested.

We comment on potential modifications that can be made to the model that may account for the discrepancy referred to by the reviewer in the discussion section: “Notably, some of the predictions of the theory did not agree with the experimental data. An increased effort cost did not accompany a reduction in the duration of harvest, and hunger did not increase saccade vigor robustly. Indeed, earlier experiments have shown that if the effort cost of harvest increases, animals who expend the effort will then linger longer to harvest more of the reward that they have earned (2). This mismatch between observed behavior and theory highlights some of the limitations of our formulation. For example, our capture rate reflected a single work-harvest period, rather than a long sequence. Moreover, the capture rate did not consider the fact that the food tube had finite capacity, beyond which the food would fall and be wasted. This constraint would discourage a policy of working more but harvesting less. Finally, if we assume that a reduced body weight is a proxy for increased subjective value of reward, it is notable that we observed a robust effect on vigor of licks, but not saccades. A more realistic capture rate formulation awaits simulations, possibly one that describes capture rate not as the ratio of two sums (sum of gains and losses with respect to sum of time), but rather the expected value of the ratio of each gain and loss with respect to time (Bateson et al., 1995 & 1996).”

Page 9, line 2-11: In this section, it would help to also consider 'baseline' pupil size (inbetween trials). This would give a signal that is not 'contaminated' by movements, and may reflect control state. Relatedly, changes in control state may impact and confound the movement-related dilation magnitudes due to e.g. floor and ceiling effects on pupil size, which has a strong tendency for reversion to the mean.

The experiment design included little or no between-trial periods because during the trials the subjects worked (performed saccades to accumulate reward), while after completing a few trials they stopped working and started harvesting through licking. Because primates make saccades during their entire wake state, it is probably not possible to find a significant period in which the subjects do not make any movements. We selected a window of 500 ms around each lick in the harvest period, and each saccade during the work period, and computed the average pupil size per movement, which includes data from both before and after movements. We then computed a within-session z-score by normalizing these measures by the average pupil size acquired for that day.

The hunger-related and reward-size related analyses are both heavily confounded since they were not manipulated directly and could co-vary with many latent factors. For example, why might a given Marmoset be lower weight on a given day? Could it affect sleep, stress, activity, or other factors during the preceding 24 hours? If so, could these other variables be driving the results that are interpreted as 'hunger?' Relatedly, since the reward size is determined by the animals behavior on each trial (how much they worked), factors (internal brain state, external noises, etc.) that alter how much they worked will influence the subsequent reward size. Therefore interpretations about reward expectancy are confounded. Both of these issues should be discussed and manipulations of them (different feeding schedules and reward size-work functions proposed, respectively).

Weight of the subjects was measured prior to the start of the experiment on each day. The natural fluctuations are typically the result of factors such as time of the experiment and corresponding weight measurement (AM vs PM) relative to the time of feeding on the previous day, day of the week of the experiment (following a weekend vs. during the week), and volume of food given during the previous day. Animals were maintained at 90% of their baseline weight during food restriction, and fluctuations typically occurred within that range (Sedaghat-Nejad et al., 2019). We used weight as a proxy for hunger, and thus value of reward, and the resulting analyses yielded results consistent with predictions made by our model, as seen in Fig. 5. Critically, other factors that may co-vary with lower weights, like those mentioned by the reviewer (sleep conditions, stress levels, and activity levels) often lead to very poor task performance by the subjects. In sharp contrast, the model predicted increased work period, and increased movement vigor for high reward value, both of which we observed when the subject’s weight was low. Thus, a low relative weight did not seem to impair performance, but rather act as a motivating factor. Subjects were closely monitored for well characterized stress-related behaviors and impaired attentive states by experimenters, veterinarian staff, and caretaker staff, and, in the event of abnormalities, were removed from food restriction and experimentation until behavior stabilized.

Effect of reward size: As you noted, we did not manipulate reward size directly. Rather, because our emphasis was on quantifying the effect of effort, the subjects received the same increment of reward per each completed trial, but on some sessions this reward was easy to harvest, while in other sessions the reward required greater effort to harvest. Because the reward amount accumulated during the work period, some harvests encountered a small amount of reward, while other harvests encountered a large amount of reward. Indeed, the amount of reward available for harvest depended linearly on the number of successful saccade trials completed during the work period. We found that the vigor of licks grew with the reward magnitude.

A major issue is a lack of alternative models. The authors seem to have constructed a particular model designed to capture the behavioral patterns they observed in the data. The model fails in some instances, as they point out. Even more importantly, there are no results or discussion about how other plausible models could or couldn't fit the data. The lack of model comparisons makes it difficult to interpret the conclusions or put the results in a broader context.

To model behavior, we chose a formulation of utility that represented a normative approach that ecologists have used to understand the decisions that animals make regarding how far to travel for food, what mode of travel to use, and how long to stay before moving on to another patch. In the model, the objective of decisions and actions is to maximize the sum of reward acquired, minus the efforts expended, divided by time. This is termed the capture rate. However, there are other models to consider, and thus we added a new section titled Model formulation and Other models of utility.

**Reviewer #2 (Public Review):**
The model proposed in the paper takes a very specific functional form that is neither motivated by the previous literature nor particularly useful for indexing the behavioral tendencies of individual monkeys (or of the same monkey in different contexts). For example, while it is clear that the saccade effort cost will need to outgrow the increase in the utility of the accumulated food for the monkey to start feeding, it is unclear why this needs to be modeled with a fixed quadratic exponent on the number of saccades? Similarly, why do licks deplete the food stash with the specific rate hard-coded in the model?

We added a section titled Model formulation and Other models of utility to better explain the rationale behind the model.

We chose this formulation of utility (Eq. 1) because it is a normative approach that ecologists have used to understand the decisions that animals make regarding how far to travel for food, what mode of travel to use, and how long to stay before moving on to another reward opportunity (Richardson and Verbeek, 1986; Stephens and Krebs, 1986; Bautista et al., 2001). In a typical formulation of the theory, the numerator represents the reward gained (in units of energy), minus the effort expended (also in units of energy), while the denominator represents the amount of time spent during that behavior. We represented this idea in Eq. (1) with saccades that produced reward accumulation, and licks that produced reward consumption. Thus, the utility that we aim to maximize is the rate of energy gained.

The specific functions that we used to represent the energy gained through reward acquisition, and the energy expended through effort expenditure, came either from experiment design, or from the measurements we have made in other experiments. We modeled reward accumulation as a linear rise in energy stored because successful saccades produced a linear increase in the food cache. We modeled consumption of the food as a hyperbolic function of the number of licks to represent the fact that as the licking bout began, each successful lick depleted the food, and thus the first few licks produced a greater amount of food consumption than the last few licks. We modeled the effort cost of licking to grow linearly with the number of licks.

A critical assumption that we made is that energy expended performing the saccade trials (which grew faster than linearly as a function of the number of trials attempted), grew faster than the time spent attempting those same trials (which grew linearly with the number of trials). This assumption is based on the heuristic that the average rate of energy lost following a large number of attempted trials is greater than the average rate of energy lost following a small number of attempted trials. A quadratic function is one example of such a function, which has the advantage of providing closed form solutions for the optimal policy.

The model’s simplicity provided closed-form solutions across all parameter values, allowing us to make predictions without having to fit the model to the measured data. Critically, for all parameter values that produce a real solution (as opposed to imaginary), the optimal number of saccade trials increases with the square root of the cost of licking. Thus, the basic prediction of the model is that to maximize the capture rate, regardless of parameter values, an increase in the effort required for harvest should be met with a greater willingness to work. The closed-form solutions are presented in the supplementary document (simulations.nb).

Finally, the proportion of successful saccades and lick events is assumed to be fixed, even though it very likely to be directly influenced by movement speed (speed- accuracy trade-off), which is also contained in the model. It would strongly increase the plausibility and potential impact of the model if the authors could clearly state where these hard-coded model terms come from. Ideally, they would formulate the model in more general terms and also consider other functional forms, as briefly suggested in the discussion. This latter point would be particularly important since not all model predictions were actually borne out in the data.

Thank you for this excellent suggestion. Regarding saccades, contrary to the speed accuracy trade-off hypothesis, we found that faster saccades were also more accurate (Fig. 3C). Thus, increased pupil size was not only associated with more vigorous saccades, but also more accurate saccades. Importantly, these vigor-related changes in accuracy were too small to affect the probability of reward: the reward area for the saccades was much larger (1.5 deg) than the endpoint accuracy changes that was produced due to changes in the food tube distance. For example, on average saccade vigor changed from 0.95 to 1.05 when the food tube distance changed from 12 mm to 8 mm. These changes in vigor would produce a fraction of degree reduction in endpoint error (Fig. 3C).

Regarding licks, we added new data to the manuscript to assess the relationship between vigor of the licks and endpoint accuracy. We saw no consistent relationship, across subjects or effort conditions, between protraction speed and the outcome of a lick, that is, if the lick was successful in making it inside the tube. On average, in subject R we observed an improvement in lick accuracy with increased vigor, and in subject M we saw no change (Fig. 4F). Thus, we used the average success rate of licks, which was roughly 30% for both subjects.

The authors derive qualitative predictions, by simulating their model with apparently arbitrary parameters. They then test these qualitative predictions with conventional statistics (e.g., t-tests of whether monkeys lick more for high vs low effort trials). The reader wonders why the authors chose this route, instead of formulating their model with flexible parameters and then fitting these to data. This would allow them (and future researchers) to test their model not just qualitatively but also quantitatively, and to compare the plausibility of different functional forms. The authors certainly have enough data and power to do this, given the vast number of sessions the monkey completed.

The model’s simplicity provides closed-form solutions across all parameter values, allowing one to make predictions without having to fit the model to the measured data. For example, for all parameter values that produce a real solution (as opposed to imaginary), the optimal number of saccade trials increases with the square root of the cost of licking. Thus, the basic prediction of the model is that to maximize the capture rate, an increase in the effort that it takes to harvest the reward should produce a greater willingness to work longer, caching more food. The closed-form solutions are presented in the Mathematica supplementary document.

The effort manipulation chosen by the authors (distance of food tube) goes hand in hand with a greater need for precision since the monkey's tongue needs to hit an opening of similar size, but now located at a greater distance. This raises the question of whether the monkeys moved slower to enhance its chance of collecting the food (in line with a speed-accuracy trade off). The manuscript would benefit from an explicit test of this possibility, for example by reporting whether for each of the two conditions, the speed of tongue movements on a trial-by-trial basis predicts the probability of food collection? At the very least, the manuscript should explicitly discuss this issue and how it affects the certainty with which effects of tube distance can be linked to anticipated effort cost alone.

Thank you for the excellent point. We looked for but found no consistent relationship, across subjects or effort conditions, between protraction speed of the tongue and the success probability of a lick (probability of insertion into the tube). Regardless, we agree with you that it is an excellent alternate hypothesis that reductions in lick vigor that accompanied increased distance of the tube may be due to a desire to maintain accuracy, and not a reflection of increased effort cost of reward. To incorporate this idea into the model, we would need a measure of speed-accuracy for the licks, something that we do not have but hope to develop in the future.

However, perhaps the most interesting aspect of our results is that when we increased tube distance, making reward more effortful, there was not only a reduction in lick vigor, but also a reduction in saccade vigor. That is, the decisions and actions during the work period responded to the increased effort cost of reward during the harvest period. These changes accompanied dilation of the pupil, both in the work period and in the harvest period. We now include a paragraph regarding this in the Discussion.

The manuscript measures pupil dilation in a time period ranging from -250ms before to 250 ms after saccade onset. However, the pupil changes strongly during saccade execution relative to the preceding baseline, leaving doubts as to whether the aggregated measure blurs several interesting and potentially different effects. It would be more conclusive if the manuscript could report the analyses of pupil size separately for a period prior to saccade onset and during/after the saccade.

Our goal was to test for general correlations between the state of the pupil and both movement vigor and decisions. We chose a window of 500 ms around saccade onset, as referred to by the reviewer, as it allowed us a large enough time window to measure pupil size outside of the movement itself (~30 ms duration), to accurately capture the state of the animal around initiation and end of a saccade. Critically, pupil tracking during a saccade itself, when using infrared eye tracking techniques, can be prone to slight measurement error in certain cases due to tracking jitter. Thus, averaging across this window, following processing of the signal, results in a more accurate measure of pupil size.